# Study on Wellbore Stability of Multilateral Wells under Seepage-Stress Coupling Condition Based on Finite Element Simulation

**Hao Xu** [1], **Jifei Cao** [1], **Leifeng Dong** [2] and **Chuanliang Yan** [2,*]

1　Drilling Technology Research Institute, SINOPEC Shengli Petroleum Engineering Co., Ltd., Dongying 257000, China; xuhaoswpu@163.com (H.X.); caojifeidrilling@163.com (J.C.)
2　Petroleum Engineering College, China University of Petroleum (East China), Qingdao 266580, China; s20020019@s.upc.edu.cn
*　Correspondence: yanchuanliang@163.com

**Abstract:** The use of multilateral wells is an important method to effectively develop complex oil reservoirs, and wellbore stability research of multilateral wells is of great importance. In the present study, the effects of formation fluids and rock damage were not taken into account by the wellbore stability model. Therefore, finite element analysis (FEA) software was used to establish a three-dimensional (3D) seepage-stress FEA model for the multilateral junctions. The model was used to analyze the wellbore stability of multilateral wells and study influences of wellbore parameters and drilling fluid density on wellbore stability at multilateral junctions. Simulation results show that the wellbore diameter insignificantly affects wellbore stability. When the angle between the main wellbore and branches enlarges to 45°, the equivalent plastic strain decreases by 0.0726, and the wellbores become more stable; when the angle is larger than or equal to 45°, the region prone to wellbore instability transfers from the multilateral junctions to the inner of multilateral wellbores. When the azimuth of wellbores is along the direction of the minimum horizontal principal stress, the equivalent plastic strain decreases by 78.2% and the wellbores are most stable. Moreover, appropriately increasing the drilling fluid density can effectively reduce the risk of wellbore instability at the multilateral junctions. A model has been developed that allows analysis of multilateral wellbore stability under seepage-stress coupling condition.

**Keywords:** multilateral well; drilling fluid; rock mechanics; pore pressure; seepage field; wellbore stability

## 1. Introduction

Wellbore instability has been one of the major technical difficulties plaguing the petroleum engineering field for a long time [1]. It may induce wellbore collapse and expansion and even severe displacement of wellbores, causing scrapping of oil and gas wells. All of these difficulties seriously affect the normal operation of oil and gas development. According to statistics, wellbore instability in the drilling process increases the drilling cost by 10~20% [2], and the time taken thereby accounts for more than 40% of the out-of-hole time [3,4]. Wellbore stability not only affects the quality of well construction engineering but is also closely related to the subsequent production or well stimulation measures [1].

Wellbore instability is a result of joint action of drilling-induced stress concentration around a well and drilling fluids, and it is a complex multi-field coupling problem [5–7]. To solve wellbore instability under multi-field coupling, Wang and Ghassemi proposed a chemo-poro-thermoelastic coupled wellbore stability model, which considers influences of temperature, seepage, and hydration on stability of shale wellbores [8,9]. Wang built a fluid-solid-chemical coupled wellbore stability model, which takes fluid flow induced by electrochemical potential and ionic diffusion and its influences on solid deformation

into consideration [10]. Zhang adopted the Weibull statistical model to describe strength deterioration of shale related to hydration strain [11]. A calculation model for collapse pressure of shale under the inhibition-blockage-seepage action of drilling fluids was also established combining with the flow-diffusion coupling model. In recent years, wellbore stability research based on multi-field coupling has expanded to bedded strata and investigates wellbore stability of bedded shale by comprehensively considering the interaction between bedding planes and drilling fluids [12,13].

With the wide application of multilateral wells and the constant increase in the drilling depth, the wellbore stability at multilateral junctions has become a key factor that influences the successful drilling and completion of multilateral wells. Failure of multilateral junctions is a serious problem. Once it is handled improperly, the wellbores will be scrapped, so the junction stability in multilateral wells has to be ensured in a bid to guarantee the smooth drilling and completion of multilateral wells [14,15]. On the one hand, sidetrack drilling of branches from the main wellbore may cause redistribution of geostress and induce stress concentration at multilateral junctions, meaning that the geometry at the junctions changes. As a result, the collapse pressure of wellbores varies, which causes wellbore instability [1]. On the other hand, because drilling fluids enter strata, the mechanical properties of rocks also change, which alters the mechanical states of rocks around the wellbore and causes instability of the wellbore system [16,17]. Therefore, higher requirements have been set for wellbore stability when using multilateral drilling technology.

Goshtasbi [18] used the finite difference method to analyze the wellbore stability of multi-lateral wells, and the results showed that the mud pressure required to stabilize the connecting section of boreholes was much greater than that required for the lateral and main boreholes. Goshtasbi [19] used the FLAC3D numerical calculation to evaluate factors such as the magnitude and distribution of ground stress and horizontal trajectory in branching Wells. The results showed that, as the dip increases, the laterals become more stable and the direction of the maximum principal stress become the optimal direction for all the stress states considered. Mohamad [20] used the hyperbolic hardening Mohr–Coulomb failure criterion to study the borehole stability of open hole multilateral Wells. The results showed that failure areas would appear at the intersection of main borehole and branch borehole, especially the collapse risk near the sharp corner of the wall would increase significantly. The branch well model established by Hoang [21] can solve the complex three-dimensional anisotropic stress state. The results show that the well inclination and azimuth of the branch well play a crucial role in the wellbore stability.

At present, there are different shortcomings in the theoretical models of wellbore stability. For example, the effect of formation fluid is not considered in the elastic model of rock. Even for the multi-field coupling models based on the energy conservation principle or stress superposition method, the established models are usually simplified, resulting in the loss of coupling meaning for some equations [22]. Some specific parameters in the models are difficult to determine in the test process, thus affecting the accuracy of the models [23,24]. Because micro-fractures in rock are the root cause of damage and destruction of rock materials, it is necessary to take rock damage into account in the process of lateral well drilling. However, so far, the traditional fluid–solid coupling model is still used in wellbore stability analysis.

In order to ascertain the influences of wellbore parameters on wellbore stability of multilateral junctions and improve the accuracy of the wellbore stability analysis, this paper introduced the plastic damage of rock material according to the theory of continuous rock damage. Finite element analysis (FEA) software was used to establish a fluid-structure coupling model for the stress-seepage field of rocks to study wellbore stability at multilateral junctions in the stress-seepage coupling field. This provides an important theoretical basis for drilling of multilateral wells in the development process of oil and gas fields.

## 2. Mathematical Model

### 2.1. Seepage Model

The seepage field of rocks changes with the stress field. According to Darcy's law, the hydraulic gradient and seepage velocity of porous media have a certain linear relationship under certain conditions, as expressed below:

$$\overline{v} = \frac{Q}{A} = -k\,grad\,H = kJ \tag{1}$$

$$H = z + \frac{p_w}{g\rho_w} \tag{2}$$

where $Q$ is the flow rate, $m^3 \cdot s^{-1}$; $A$ is the cross-sectional area of flow, $m^2$; $grad\,H$ is the hydraulic gradient; $z$ is the height above a reference surface, m; $\rho_w$ is the fluid density, $kg \cdot m^{-3}$; $H$ is the piezometric head, m; $k$ is the permeability, $m \cdot s^{-1}$; and $\overline{v}$ is the average seepage velocity, $m \cdot s^{-1}$.

### 2.2. Effective Stress Model

Rocks in a formation are mainly composed of the solid particle skeleton, pores in the skeleton, and fluids in pores. When external loads are applied to rocks in a formation, the solid particle skeleton will be partially deformed, thus inducing changes in permeability and porosity of rocks. After deformation of the solid particle skeleton, the flow state and fluid pressure in the pores also vary accordingly [25].

The tensile stress on rocks is defined as a positive by using the Biot effective stress method. In addition, it is determined that the pore pressures in the unsaturated and saturated regions are separately a negative and a positive. The Biot effective stress is expressed as [26]:

$$\sigma'_{ij} = \sigma_{ij} - \alpha P_p \delta_{ij} \tag{3}$$

where $\sigma_{ij}$ and $\sigma'_{ij}$ separately represent total stress component and effective stress components of rocks around wellbores, MPa; $P_P$ is the pore pressure, MPa; and $\alpha$ is the coefficient of effective stress and is defined as follows:

$$\alpha = 1 - \frac{K_V}{K_S} \tag{4}$$

where $K_S$ is the compression modulus of solid particles, MPa; and $K_V$ is the volume compression modulus of rocks, MPa.

### 2.3. Equilibrium Equation

The stress equilibrium equation is expressed according to the principle of virtual work. That is, within the same time period, the virtual work performed by rock and soil mass is identical to the virtual work produced by forces acting on rock and soil mass, as expressed below [27]:

$$\int_V \delta\varepsilon^T d\sigma dV - \int_V \delta u^T df dV - \int_S \delta u^T dt dS = 0 \tag{5}$$

where $\delta u$ is the virtual displacement, m; $\delta\varepsilon$ is the virtual strain; $t$ is the surface force, N; and $f$ is the body force, N.

The constitutive relationship is expressed as follows:

$$d\overline{\sigma} = D_{ep}\left(d\varepsilon - d\varepsilon_f\right) \tag{6}$$

where $D_{ep}$ is the elastic-plastic matrix and $d\varepsilon_f$ is the particle compression induced by pore pressure, and it is expressed by:

$$d\varepsilon_f = -m\frac{d\overline{p}}{3K} \tag{7}$$

where $m = [1, 1, 1, 0, 0, 0]^T$.

*2.4. Continuity Equation*

According to the principle of mass conservation, the increase in the moisture content of rocks within time $dt$ should be equal to the water content flowing in rocks in the time. Through calculation, the continuity equation for seepage in rocks is shown as follows:

$$
s_w \left( m^T - \frac{m^T D_{ep}}{3K_S} \right) \frac{d\varepsilon}{dt} - \nabla^T \left[ k_0 k_r \left( \frac{\nabla p_w}{\rho_w} \right) - g \right] +
$$
$$
\left\{ \zeta n + n \frac{s_w}{K_w} + s_w \left[ \frac{1-n}{3K_S} - \frac{m^T D_{ep} m}{(3K_S)^2} \right] (s_w + p_w \zeta) \right\} \frac{dp_w}{dt} = 0
\tag{8}
$$

*2.5. Yield Criterion*

Wellbore instability in the drilling process occurs because the shear force on rocks at wellbores exceeds their shear strength, so the Mohr–Coulomb criterion is selected as the strength criterion of the model. The Mohr–Coulomb criterion reflects the relationship between shear stress and normal stress at failure of rocks and uses the maximum and minimum principal stresses to separately represent the normal stress and shear stress. Under the pore pressure in the formation, the Mohr–Coulomb criterion is rewritten as [28]:

$$
f = (\sigma_1 - \sigma_3) - \sin \varphi (\sigma_1 - \sigma_3 - 2\eta P) - 2C \cos \varphi = 0
\tag{9}
$$

*2.6. Boundary Conditions*

Assuming that the displacement components at boundaries infinite to a wellbore are all 0, then the following is obtained:

$$
u(x_a, y_a, z_a) = u_a = 0
\tag{10}
$$

$$
v(x_b, y_b, z_b) = v_b = 0
\tag{11}
$$

$$
w(x_c, y_c, z_c) = w = 0
\tag{12}
$$

The boundary condition on wellbores is the first-order derivative of displacement:

$$
E \nabla u(x, y, z) = p_w
\tag{13}
$$

$$
E \nabla v(x, y, z) = p_w
\tag{14}
$$

$$
E \nabla w(x, y, z) = p_w
\tag{15}
$$

where $p_w$ is wellbore pressure, MPa.

**3. Numerical Model**

During stability analysis in the vicinity of a wellbore, a straight well can be solved using two-dimensional (2D) planar stress or strain. However, for multilateral wells, this may cause certain errors. Therefore, when establishing the mechanical model, influences of geometric parameters on multilateral wells were fully considered, thus building the 3D symmetric analysis model of wellbore stability. The ideal elastic–plastic constitutive model was used and the Mohr–Coulomb criterion was adopted as the criterion for wellbore instability. Figure 1 shows the 3D geometric model of multilateral wells under stress-seepage conditions and mesh generation of the model. The diameter of the main wellbore was 0.32 m, and that of branches was 0.22 m. Triangular and tetrahedral grids were used for the 3D model. To improve the accuracy of calculation results, grids in the region around

the wellbores were refined. The model is composed of 35,069 elements, in which C3D20RP is selected as the element type. The simulation requirements can be met by a general simulation computer.

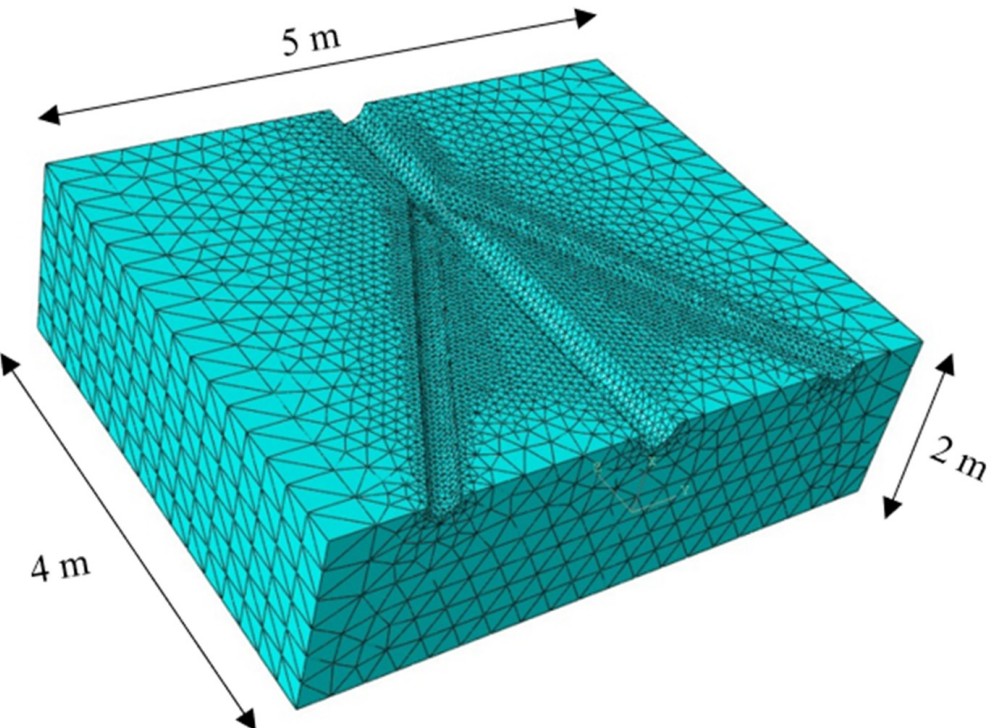

**Figure 1.** Geometric model and generated mesh.

The model belongs to a plane-symmetric model in terms of the geometric structure. According to the symmetry principle, the model could be simplified and half of the model was taken as the research object, which not only reflects the accuracy of calculation results of the model but also reduces the workload of the entire simulation process.

Parameters of the rocks and fluids in the model analysis process are listed in Table 1.

**Table 1.** Basic parameters in model analysis.

| Parameters | Values | Parameters | Values |
|---|---|---|---|
| Rock density | 2300 kg/m$^3$ | Well depth | 1300 m |
| Elastic modulus | 6000 MPa | Overburden pressure | 30 Mpa |
| Poisson's ratio | 0.25 | Maximum horizontal principal stress | 27 Mpa |
| Internal frictional force | 32° | Minimum horizontal principal stress | 23 Mpa |
| Cohesion | 5 Mpa | Formation pressure | 12.6 Mpa |
| Drilling fluid density | 1.1 g/cm$^3$ | Porosity ratio | 0.5 |

## 4. Wellbore Stability of Multilateral Wells

Figure 2 illustrates the nephogram for pore-pressure distribution in the formation under seepage conditions. It can be seen that the fluid column pressure of drilling fluids exerts tremendous influences on the pore pressure in the formation around the wellbores. Drilling fluids also may flow into the formation around wellbores through micro-fractures and pores, thus changing the stress state around the wellbores. As the distance from the wellbores enlarges, the influences of seepage constantly weaken.

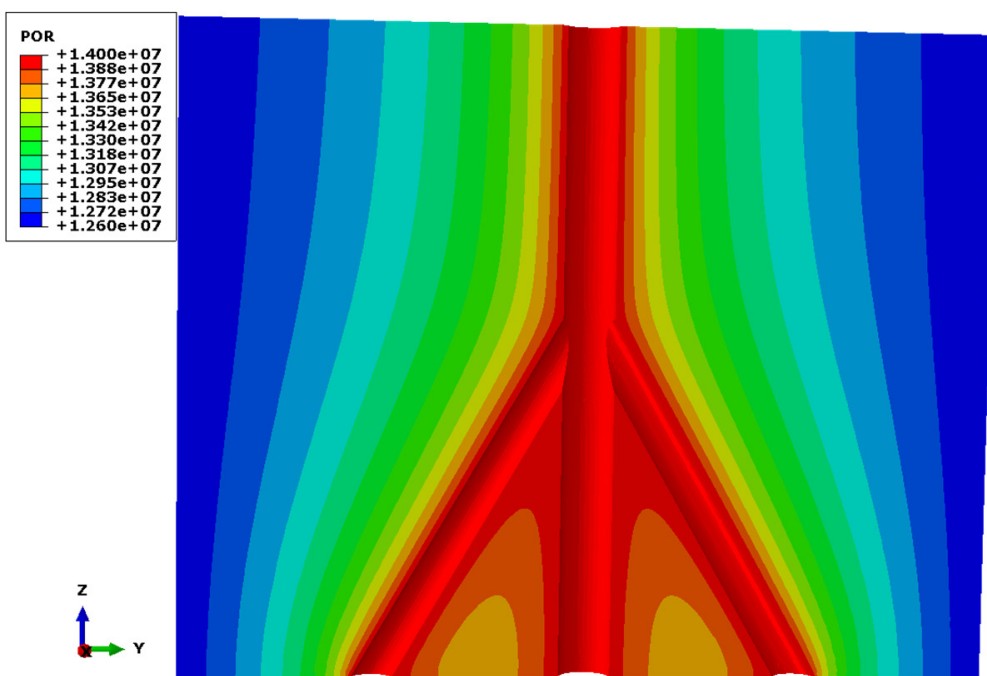

**Figure 2.** Nephogram for pore-pressure distribution under seepage, Pa.

Figure 3 shows the nephogram for the equivalent plastic strain around multilateral wells under different drilling fluid densities and considering seepage of drilling fluids. The regions with equivalent plastic strain around the wellbores are all concentrated at the multilateral junctions. As the drilling fluid density rises, the equivalent plastic strain decreases. This is because, the higher the drilling fluid density is, the greater the fluid column pressure of drilling fluids in the wellbores, the larger the support for wellbores, and the lower the stress difference at multilateral junctions. Therefore, the value and the area of the regions with equivalent plastic strain both reduce constantly until they disappear. Increasing the drilling fluid density is favorable for the wellbore stability of multilateral wells; at the same time, it increases the risk of leakage or differential pipe sticking and also lowers the rate of penetration. Therefore, multiple factors should be considered comprehensively to effectively reduce the instability risk at multilateral junctions by appropriately increasing the drilling fluid density.

To analyze influences of seepage of drilling fluids on wellbore stability of multilateral wells, the wellbore stability of multilateral wells not considering seepage of drilling fluids was calculated. Figure 4 shows the change curves of the maximum equivalent plastic strain with the drilling fluid density. It can be seen that, with the increase in the drilling fluid density, the maximum equivalent plastic strain formed around the wellbores reduces either the under seepage or non-seepage condition. Under the stress-seepage field, the decrease rate of equivalent plastic strain under seepage is lower than that under the non-seepage condition with the increasing drilling fluid density. This is because additional stress is generated when fluids enter the formation from wellbores in the seepage process. Under this condition, to ensure stress in the formation to reach equilibrium, the drilling fluid density needs to be increased to offset such additional force. Hence, wellbores of multilateral wells can find equilibrium more rapidly under the non-seepage condition with the increase in drilling fluid density compared with that under the seepage condition.

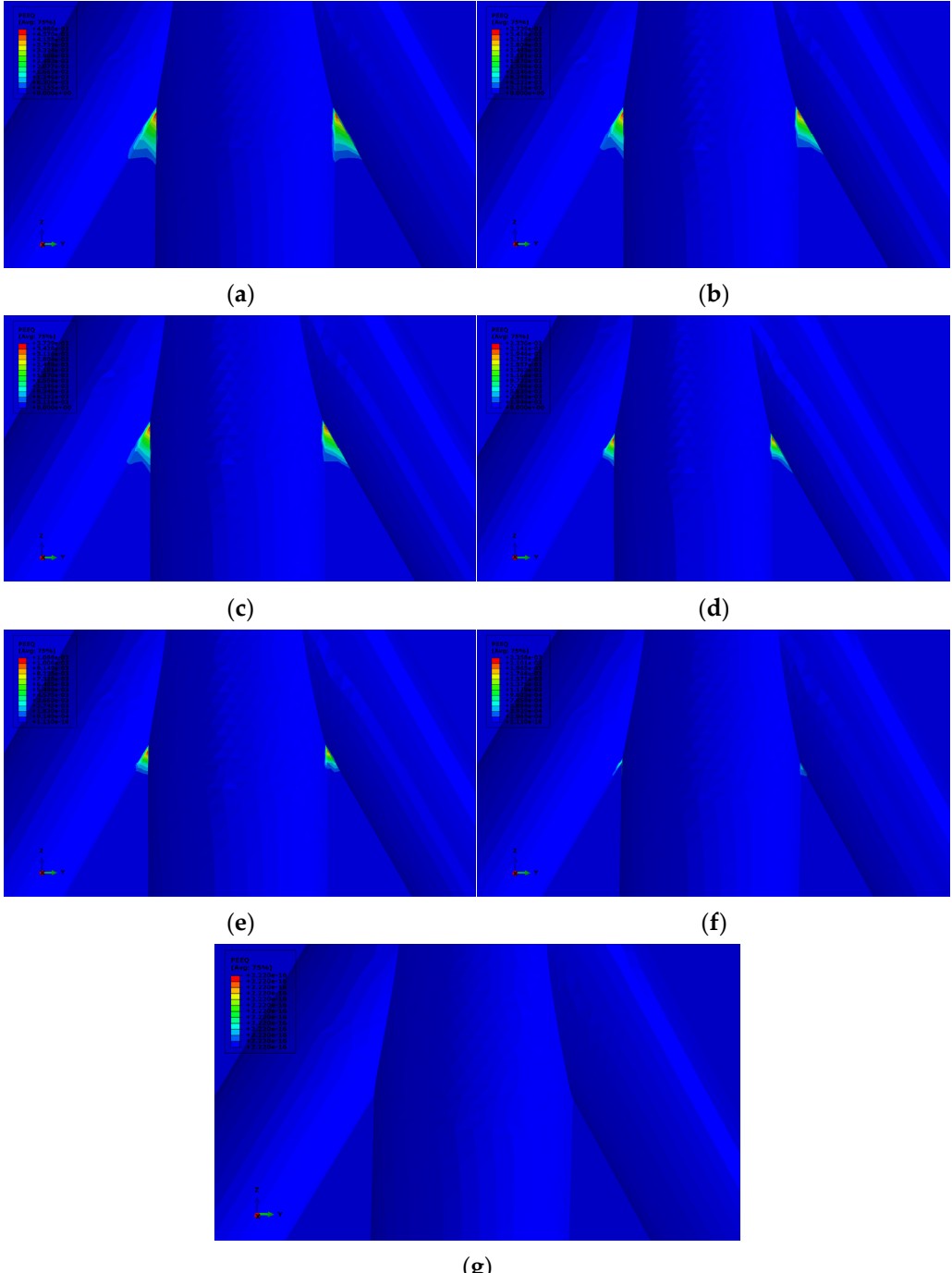

**Figure 3.** Nephogram for equivalent plastic strain under different drilling fluid densities during seepage. (**a**) 0.9 g/cm³, (**b**) 1.0 g/cm³, (**c**) 1.05 g/cm³, (**d**) 1.1 g/cm³, (**e**) 1.2 g/cm³, (**f**) 1.3 g/cm³, (**g**) 1.33 g/cm³.

The two curves under seepage and non-seepage conditions intersect at a point where the drilling fluid density is about 1 g/cm³. When the drilling fluid density is lower than the value at the intersection, the maximum equivalent plastic strain under the non-seepage condition is larger than that under the seepage condition. If the drilling fluid density exceeds the value at the intersection, the maximum equivalent plastic strain under the non-seepage condition is lower than that under the seepage condition. Because the formation pressure is 12.6 MPa, the two curves under seepage and non-seepage conditions are intersected when the fluid column pressure of drilling fluids is 12.6 MPa. In the FEA

model, the boundary conditions of pore pressure under the two conditions are the same and the fluid column pressure of drilling fluids in wellbores is identical to the formation pressure, so the calculated maximum equivalent plastic strain is also same.

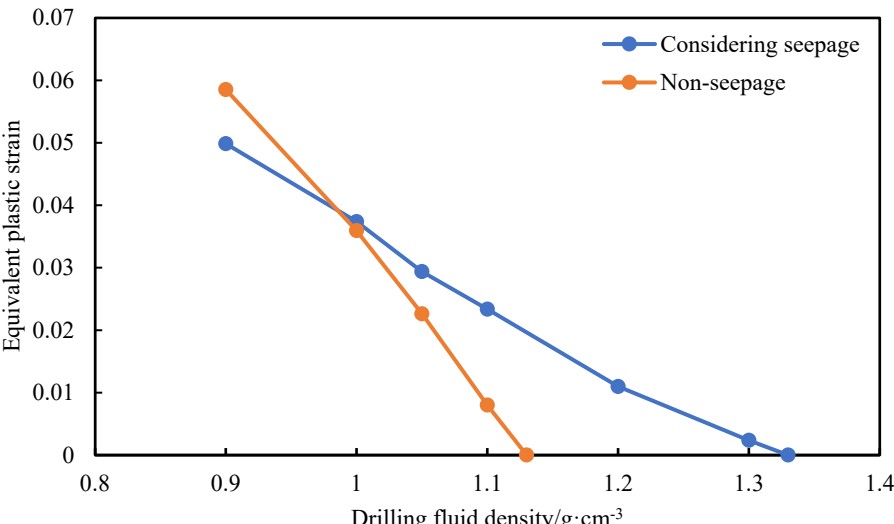

**Figure 4.** Change trends of the equivalent plastic strain with different drilling fluid densities under seepage and non-seepage conditions.

In the drilling process, to avoid blowout, the fluid column pressure of drilling fluids should be larger than the pore pressure. The equivalent plastic strain under the seepage condition is greater than that under the non-seepage condition. As the drilling fluid density grows, the difference between the two becomes increasingly larger. Therefore, seepage exerts remarkable influences on the wellbore stability of multilateral wells and is indispensable in the analysis of wellbore stability of multilateral wells.

As shown in Figure 5, the effective stress in the formation as well as the maximum and minimum principal stresses are lower under the seepage condition during drilling. Therefore, the Mohr's circles of stress at the multilateral well junction is close to the failure envelope (blue dotted line), and the wellbores become more unstable.

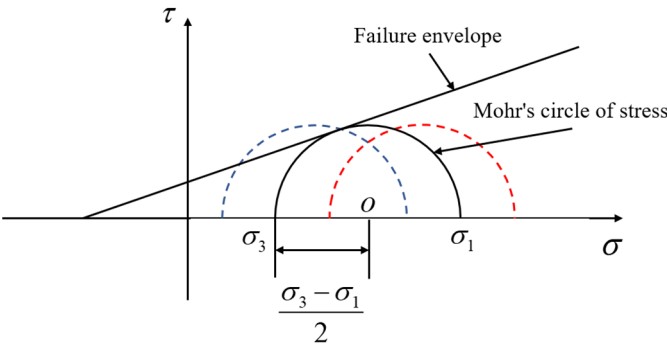

**Figure 5.** Schematic diagram for the Mohr's circles of stress.

Figure 6 illustrates the change trends of the minimum and maximum principal stresses under different drilling fluid densities. It can be seen that the minimum and maximum principal stresses intersect at the drilling fluid density of 1.0 g/cm$^3$ under seepage and non-seepage conditions. In the drilling process, the maximum and minimum principal stresses under the seepage condition are lower than those under the non-seepage condition. So, the Mohr stress circle shifts leftwards to approach the failure envelope in the process and the wellbores are more prone to failure.

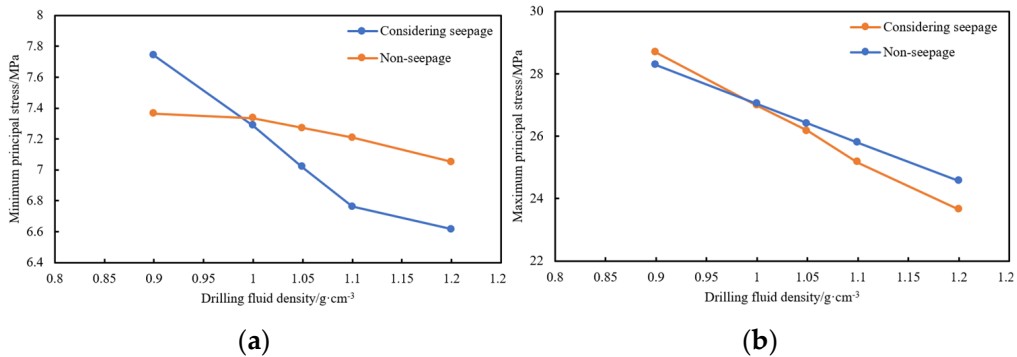

(**a**)                      (**b**)

**Figure 6.** Change trends of the principle stress under different drilling fluid densities. (**a**) minimum principle stress, (**b**) maximum principle stress.

## 5. Influencing Factors of Wellbore Stability of Multilateral Wells

FEA software was used to change different wellbore parameters in order to investigate wellbore stability at the multilateral junctions and analyze influences of various factors.

### 5.1. Influences of Wellbore Diameters of Multilateral Wells

Figure 7 illustrates the nephogram for equivalent plastic strain under different wellbore diameters of multilateral wells. It can be seen from the figure that the regions with equivalent plastic strain are all concentrated at the multilateral junctions. Stress is concentrated at the junctions between the main wellbore and branches. As the wellbore diameter of branches enlarge, the value and the range of regions with the equivalent plastic strain both reduce. The wellbore diameter of multilateral wells exerts slight influences on the wellbore stability.

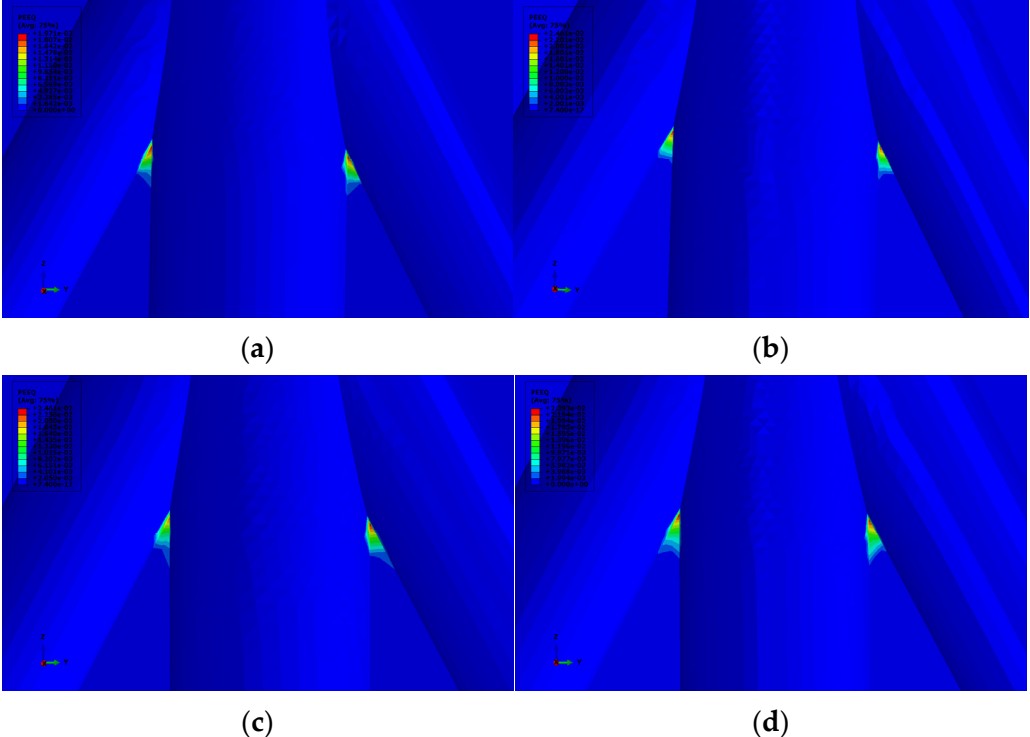

(**a**)                      (**b**)

(**c**)                      (**d**)

**Figure 7.** *Cont.*

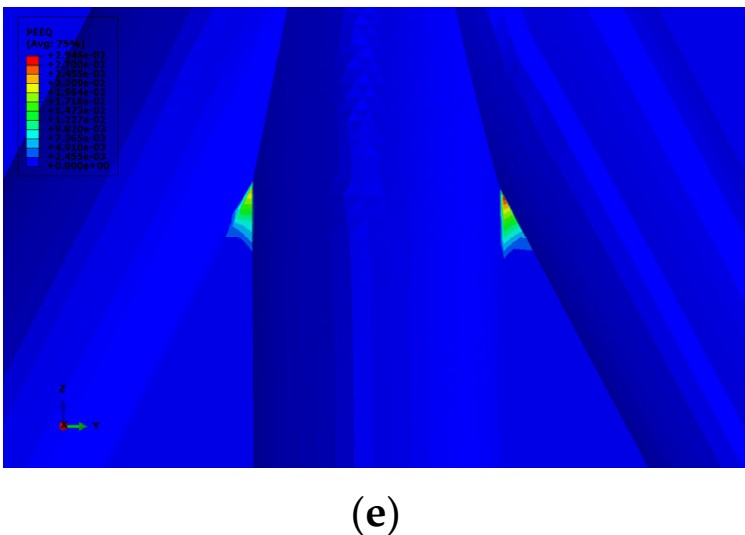

(**e**)

**Figure 7.** Nephogram for the equivalent plastic strain under different wellbore diameters of multilateral wells. (**a**) Wellbore diameter of 0.20 m, (**b**) Wellbore diameter of 0.22 m, (**c**) Wellbore diameter of 0.24 m, (**d**) Wellbore diameter of 0.26 m, (**e**) Wellbore diameter of 0.28 m.

Figure 8 shows changes in the maximum equivalent plastic strain of different multilateral wells with the wellbore diameter. It can be seen from the figure that, with the enlargement of the wellbore diameter of multilateral wells, the maximum equivalent plastic strain at multilateral junctions increases slightly while its value varies slightly. This reveals that, although enlarging the diameter of multilateral wells may increase the risk of wellbore instability, it exerts limited influences on the mechanical stability of multilateral junctions.

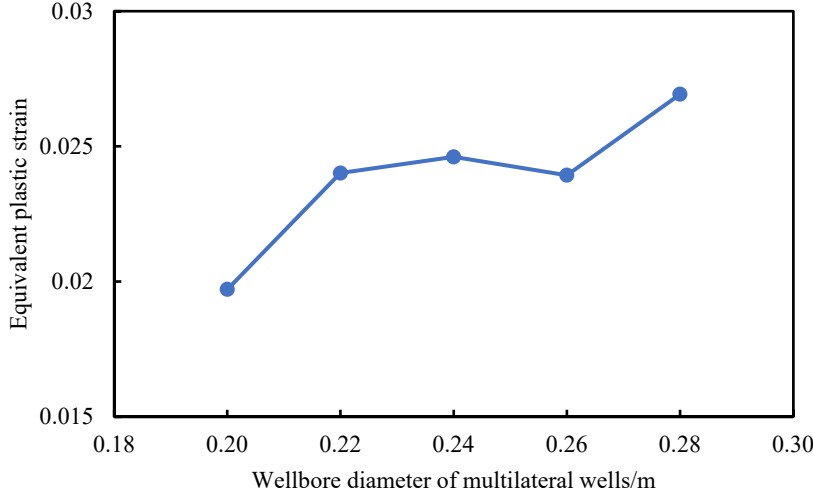

**Figure 8.** Change trend of the maximum equivalent plastic strain under different wellbore diameters of multilateral wells.

### 5.2. Influences of the Angle between Main Wellbore and Branches

Figure 9 shows nephograms for Mises stress distribution when the angles between main wellbore and branches are 30° and 45°. When the angle is smaller than 45°, the multilateral junctions are likely to be damaged due to stress concentration, while the stress difference in multilateral wellbores is small and the wellbores are not damaged. If the angle is larger than or equal to 45°, the maximum stress transfers from the multilateral junctions to the inner of wellbores. Under the condition, the stress difference in multilateral wellbores begins to increase and the regions with equivalent plastic strain start to transfer from the multilateral junction to the inner of multilateral wellbores.

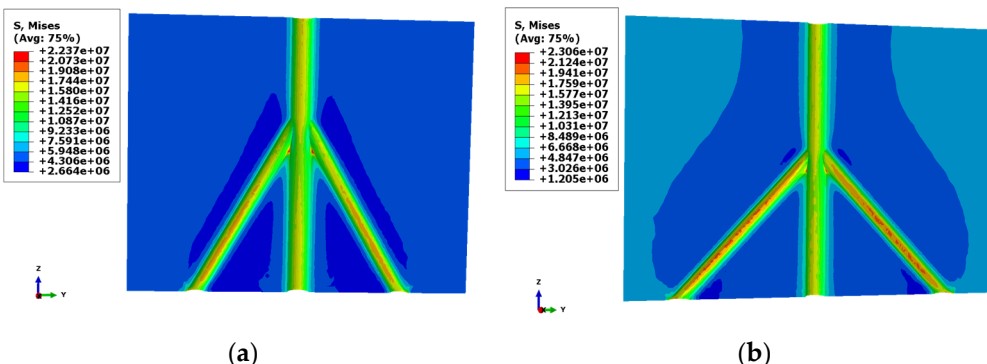

**Figure 9.** Nephogram for Mises stress distribution, Pa. (**a**) The angle between main wellbore and branches is 30°, (**b**) The angle is 45°.

The nephograms for distribution of the equivalent plastic strain under different angles between main wellbore and branches are illustrated in Figure 10. It can be seen that, when the angles are 15° and 30°, the equivalent plastic strain in the wellbores is mainly concentrated at multilateral junctions. It has a large value, and the wellbores in the region with equivalent plastic strain are more prone to failure. When the angle exceeds 45°, the equivalent plastic strain in the wellbores begins to transfer from the multilateral junction to the inner of multilateral wellbores. This is because wellbore structure is special at the multilateral junctions when the angle is small. At the location, stress concentration is likely to occur while the stress difference in multilateral wellbores is so small that no stress concentration appears. When the angle is large, the stress difference in the multilateral borewells begins to enlarge and stress concentration starts to appear in the multilateral wellbores. At the same time, stress concentration at the multilateral junctions gradually weakens and disappears, and the regions with equivalent plastic begin to transfer from the multilateral junctions to the surrounding areas of multilateral wellbores.

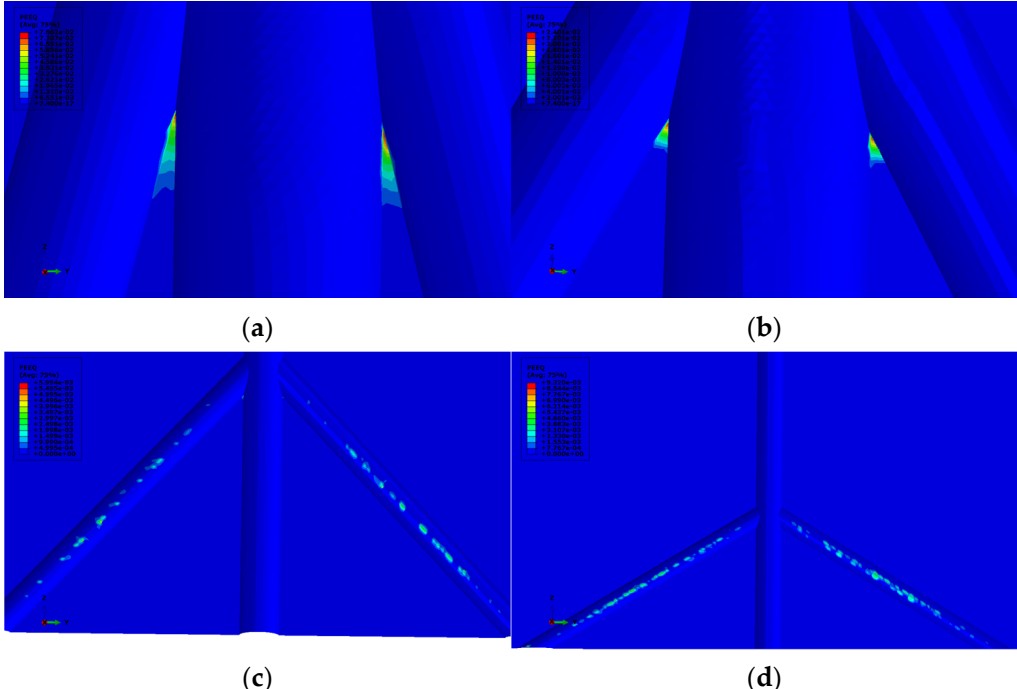

**Figure 10.** *Cont.*

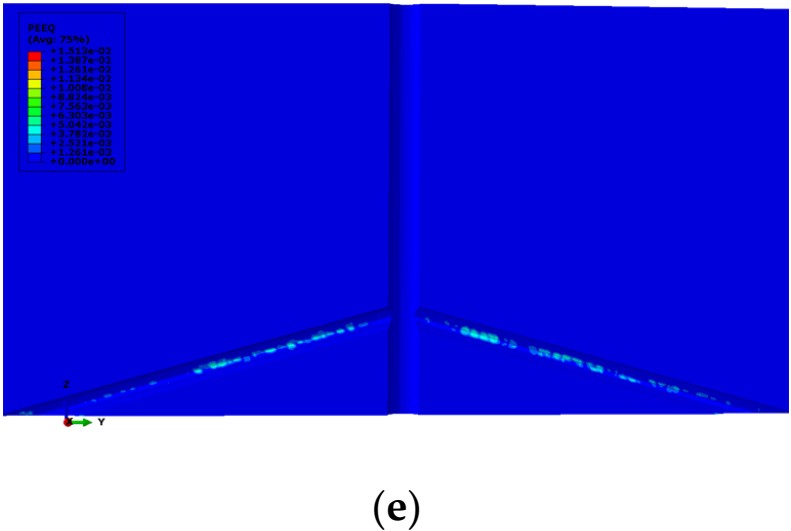

**(e)**

**Figure 10.** Nephograms for distribution of the equivalent plastic strain under different angles between main wellbore and branches. (**a**) 15°, (**b**) 30°, (**c**) 45°, (**d**) 60°, (**e**) 75°.

Figure 11 displays the change trend of equivalent plastic strain under different angles between the main wellbore and branches. As the angle enlarges, the maximum equivalent plastic strain decreases quickly at first, and then more slowly when the angle is smaller than 45°. The equivalent plastic strain rises slowly with the angle when the angle exceeds 45°; in the process, the regions of wellbore instability transfer from the multilateral well junction to the inner areas of multilateral wellbores. This is because, with the enlargement of the angle between main wellbore and branches, the equivalent plastic strain only appears in the multilateral wellbores. This can be approximately regarded as changes in wellbore stability of common directional wells: the larger the wellbore angle is, the greater the stress difference in wellbores, and therefore, the larger the equivalent plastic strain.

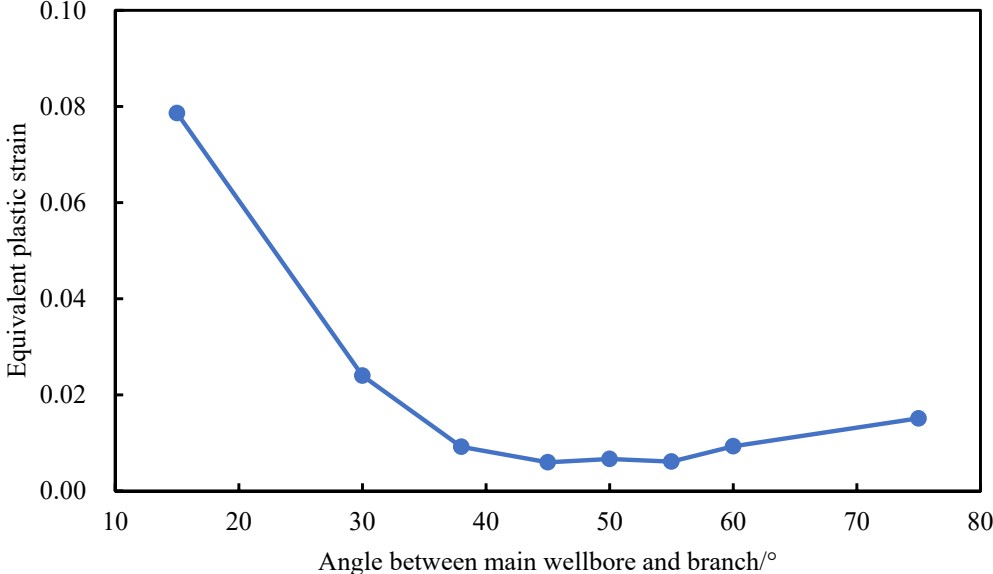

**Figure 11.** Change trend of the maximum equivalent plastic strain with different angles between main wellbore and branches.

*5.3. Influences of Azimuth of Multilateral Wells*

Figure 12 show the nephograms for Mises stress distribution when the relative azimuths (angles with the orientation of maximum horizontal geostress) of multilateral wells

are separately 0° and 90°. It can be seen from the figures that, although Mises stress declines with the enlargement of the relative azimuth from 0° to 90°, the maximum Mises stress is always found at the multilateral junctions. This explains why the stress concentration at multilateral junctions is most serious. Figure 13 shows the change trend of Mises stress under different azimuths. As the azimuth enlarges, the maximum Mises stress approximately linearly decreases, which indicates that stress concentration at multilateral junctions also weakens with the enlargement of azimuth and the wellbore stability is also enhanced.

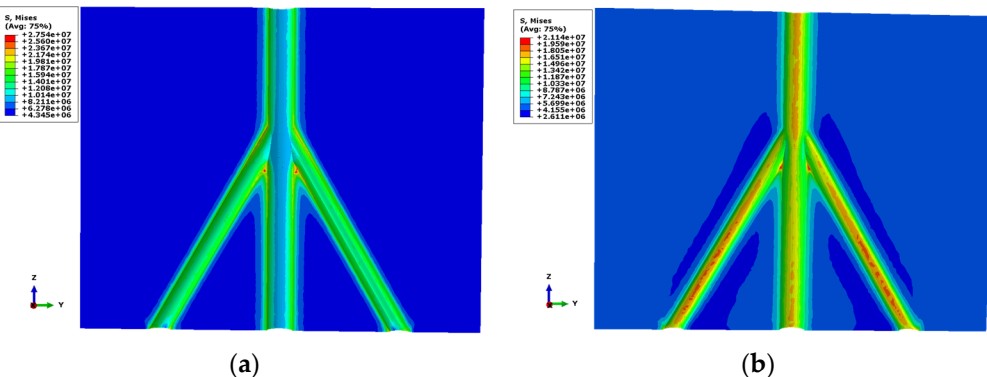

|           |           |
|-----------|-----------|
| (**a**)   | (**b**)   |

**Figure 12.** Nephogram for Mises stress distribution, Pa. (**a**)The relative azimuth is 0°, (**b**) The relative azimuth is 90°.

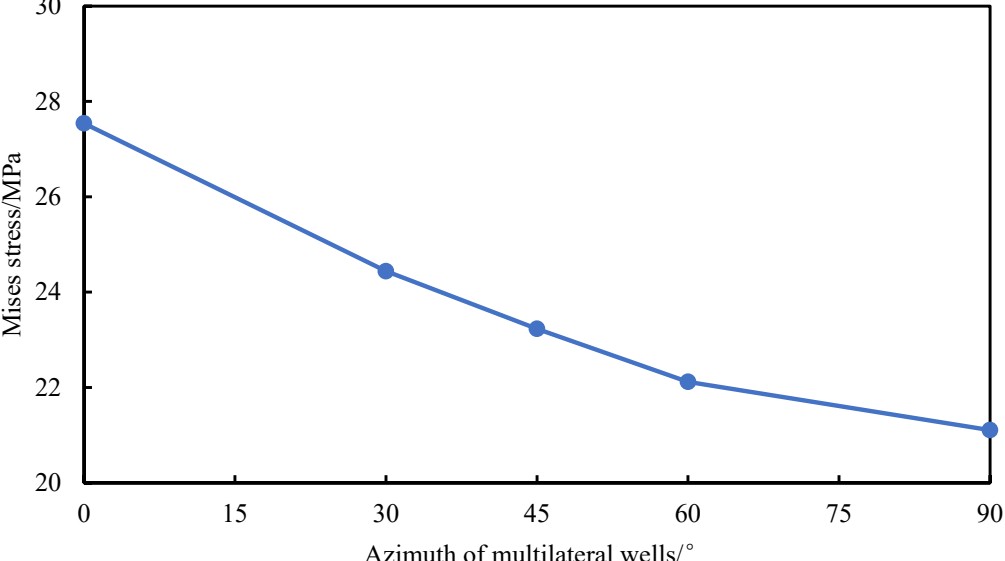

**Figure 13.** Change trend of Mises stress under different relative azimuths.

Figure 14 shows nephograms for the equivalent plastic strain under different relative azimuths of multilateral wells. As shown in the figure, the regions with equivalent plastic strain are all concentrated at multilateral junctions. These regions gradually shrink and the maximum equivalent plastic strain gradually reduces with the enlargement of azimuth of multilateral wells. When the relative azimuth in the model enlarges in the range of 0~90°, the area of regions with equivalent plastic strain narrows and the wellbores become more stable.

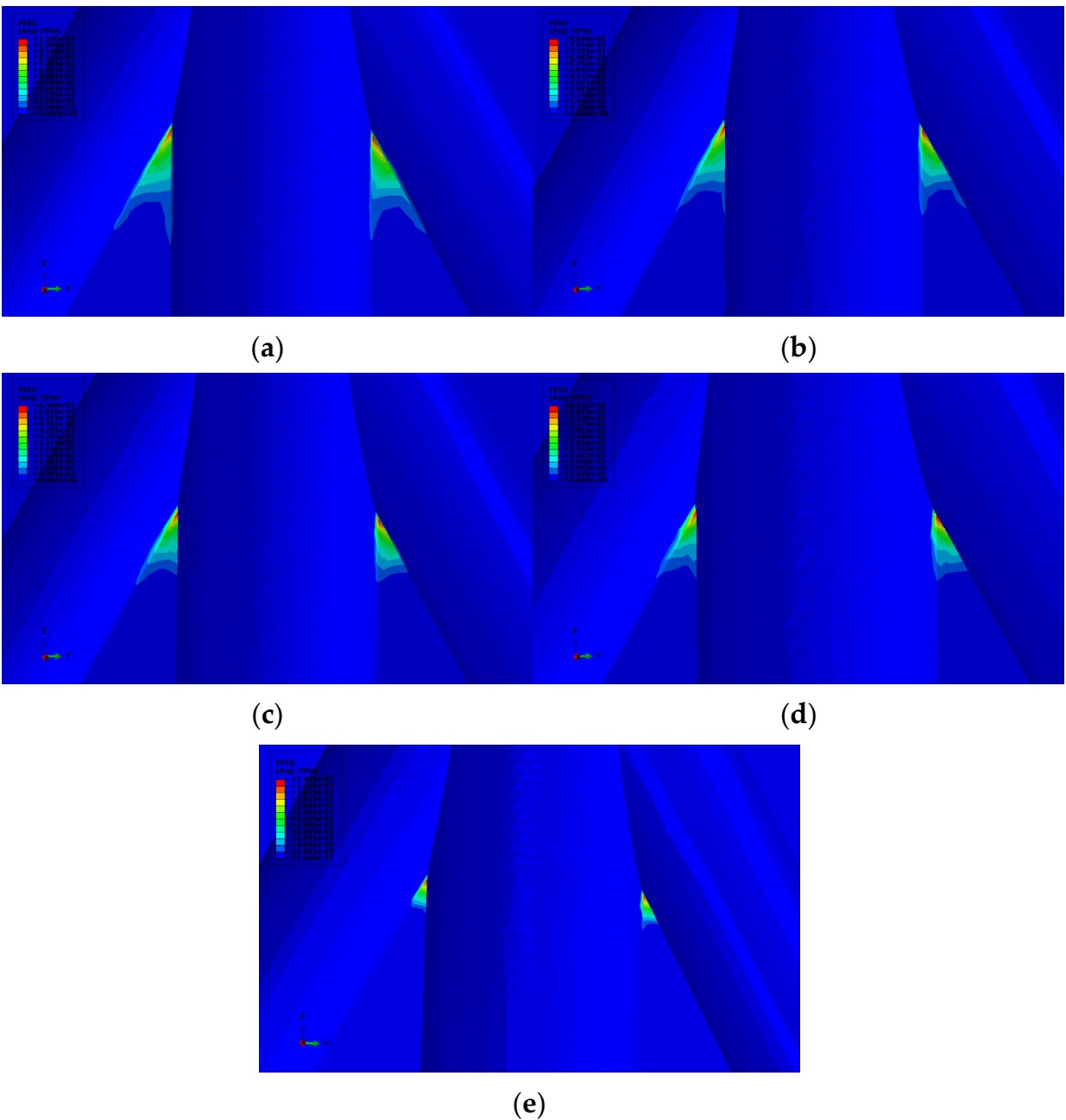

**Figure 14.** Nephograms for equivalent plastic strain under different azimuths of multilateral wells. (**a**) Azimuth of multilateral wells of 0°, (**b**) Azimuth of multilateral wells of 30°, (**c**) Azimuth of multilateral wells of 45°, (**d**) Azimuth of multilateral wells of 60°, (**e**) Azimuth of multilateral wells of 90°.

Figure 15 shows the change trend of the maximum equivalent plastic strain of wellbores with relative azimuth of multilateral wells. It can be seen from the figure that, as the relative azimuth enlarges, the equivalent plastic strain at multilateral junctions reduces quickly at first and then slowly, and the wellbores become more stable. When the azimuth approaches 0°, that is, the relative azimuth of wellbores is closer to the direction of the maximum horizontal principal stress, the equivalent plastic strain is greater and the wellbores become more unstable. When the relative azimuth is about 90°, the azimuth of wellbores is closer to the direction of the minimum principal stress, the equivalent plastic strain is lower, and the wellbores are more stable.

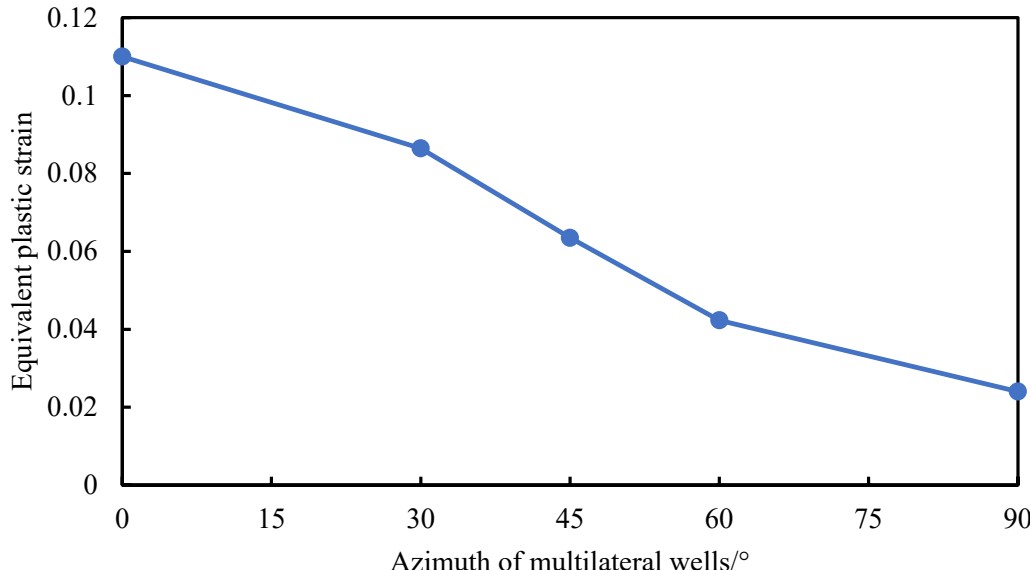

**Figure 15.** Change trend of the maximum equivalent plastic strain under different azimuths of multilateral wells.

## 6. Conclusions

In order to efficiently develop complex oil reservoirs and accurately analyze wellbore stability at multilateral junctions of multilateral wells, the finite element analysis software was used to establish a three-dimensional seepage-stress FEA model for the multilateral junctions. The simulation results show that:

(1) Stress concentration is most serious at multilateral junctions of multilateral wells, where wellbore instability is most likely to occur.
(2) The maximum plastic strain at multilateral junctions increases slightly with the enlargement of wellbore diameter of multilateral wells, and the wellbore diameter exerts slight influences of the wellbore stability.
(3) The larger the angle between main wellbore and branches, the more stable the multilateral wells. When the azimuth of multilateral wells is parallel to the direction of the minimum horizontal principal stress, the equivalent plastic strain is lowest and wellbores are most stable.
(4) Appropriately increasing the drilling fluid density can effectively reduce the risk of wellbore instability at multilateral junctions.
(5) When the angle between main wellbore and branches is larger than or equal to 45°, the regions at the risk of wellbore instability transfer from multilateral junctions to the inner areas of multilateral wellbores.

**Author Contributions:** Methodology, Conceptualization, Supervision, H.X. and J.C.; Software, Investigation, Formal analysis, H.X., J.C. and C.Y.; Writing—Review and Editing, L.D. All authors contributed critically to draft revision. All authors have read and agreed to the published version of the manuscript.

**Funding:** This research was funded by the Sinopec Research Project (P21070-2), the National Natural Science Foundation Project of China (U1762216) and the Key Research and Development Program of Shandong Province (2019GGX103025).

**Conflicts of Interest:** The authors declare no conflict of interest.

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
