# Peer review of "Study on Wellbore Stability of Multilateral Wells under Seepage-Stress Coupling Condition Based on Finite Element Simulation"

_processes, doi:10.3390/pr11061651_

Round 1

Reviewer 1 Report

The manuscript represents the numerical investigations of wellbore stability of multilateral wells. The effects of the wellbore parameters and drilling fluid density on wellbore stability at multilateral junctions. The results of the manuscript can be a useful reference for engineering design. The details of comments are as follows:

Q1. The authors only explained the general trend in the abstract. The accurate results should be added to the abstract.
Q2. The authors should specify the unit of the parameter in the formula.

Q3. The authors should specify the unit in the legend of the contour distribution. e.g. Fig. 2, 3, 8, 10, 11, 12, 14,15,17.

Q4.The authors only described the general trend of the graphs.It would be even better if the discussion and analysis about the results is added.

Reviewer 2 Report

The manuscript titled "Numerical Simulation of Wellbore Stability of Multilateral..." by Hao Xu and others is a good piece of publication. Apart from minor corrections, I see no major problems in publishing this work.

Below are my comments and more suggestions.

1. The first two sentences of the abstract should be corrected.

2. Is this keyword needed: wellbore stability?

3. First of all, the last paragraph of the introduction is definitely to be rewritten, please clearly indicate what is the topic and, above all, the purpose of the work.

4. Apart from the above, the introduction is a bit incomplete. Please complete it, and add many more citations on the net on similar topics, there is probably enough literature.

5. Materials and methods are very well and sufficiently described.

6. Fig 1 - it's quality in terms of graphics seems insufficient.

7. Fig 2 - the same - can't you improve the quality of the graphics a bit? Also, you can hardly see the X-axis selected. How long is she even there?

8. Fig 3 - also very poor quality. The axis is not visible and the graphics are blurry.

9. "It can be seen that with the increase in the drilling 186 fluid density, the maximum equivalent ..." is it that important? Would that actually seem logical?

10. "Under the condition, to ensure stress in the formation to reach 192 equilibrium, the drilling fluid density needs to be increased to offset such additional force...." how can this problem be solved? Might be worth mentioning at work.

11. Fig 4 and 18 - graphs end on the x-axis. Please correct it in the sense of adding some value there. Because it doesn't look elegant. For example in Figs 6 and 7.

12. Fig 8 for improvement.

13. Fig 9 - too much free space.

14. Figs 10 and 11 and 12 - the x-axis is not visible. As mentioned above many times.

15. Fig 12 17 - very poor quality.

16. Please complete the literature.

Overall a very good job but very poorly done.

Reviewer 3 Report

The paper considers stress distribution around multilateral well bores, studied by finite-element analysis. The physical formulation is sound and based on well-established models. Results are obtained for various influencing factors, the principal aim was to analyze the wellbore stability. The conclusions are justified by the performed analysis.

Generally, the paper is well-written, its contents will be of definite interest to specialists in rock mechanics and drilling technologies. I believe the paper can be published in Processes "as is", however, with one observation:

The software used by authors is not described adequately, it is just referred to as "FEA software". Was this a software developed by authors? Or, alternatively, some commercial package was used? What were the finite element types, mesh parameters (number of elements etc), computer resources necessary for the analysis? How the software and results obtained were validated?

I believe this information must be included in paper to substantiate the numerical results presented.

Also, on page 5 the caption "Geometric model and mesh generation" should be changed to "Geometric model and generated mesh" or similar.

Reviewer 4 Report

Manuscript "Numerical Simulation of Wellbore Stability of Multilateral Wells" by Hao Xu, Jifei Cao, Leifeng Dong, Chuanliang Yan has been submitted for review. 

I have read this manuscript with great interest. The authors addressed a rather urgent problem of improving the overall development benefit of an oil field using the finite element analysis software was used to establish a three-dimensional 3D seepage-stress FEA model for the multilateral junctions.

The manuscript has a number of significant flaws that need to be corrected.  Correcting the deficiencies listed below is necessary to improve the quality of the manuscript, enhance the ease of perception of the material presented, and increase the interest of the reader.

1) From my point of view the title of the manuscript is a bit vague and does not reflect the essence of the research. For example: Establishment of 3D seepage-stress FEA model for studying influences of wellbore parameters and drilling fluid density on wellbore stability at multilateral junctions.  

2) The abstract is not quite right. It is very vague and wrongly formed. It seems that the authors have taken some phrases out of the context of their research and put them in the abstract. The abstract should clearly state the purpose of the research and its importance to society (that is, to characterize the problem), identify the research methods and materials, and formulate conclusions clearly and concisely. The abstract lacks a "starting point," that is, information about previous research (one sentence is sufficient). From my point of view, in the abstract, such information begins with the statement: "Previous research has established that ....".  

2.1) It is desirable to avoid narrative text in the abstract. 

2.2) Try to use words and phrases: analyzed; conducted; studied; developed; proposed; established and others. It is desirable to begin sentences in the abstract with these words and phrases.

2.3) At the end of the abstract it is necessary to specify the final result obtained by the authors, for example: A model has been developed that allows ...; A dependence has been established that is ...; An effective system (technology) has been proposed, and so on.

3) From my point of view, very few keywords. In addition, keywords should be more direct and related to the content of the manuscript. 

Keywords allow the reader to quickly search for the necessary material, and the author the opportunity to popularize his research, as well as to increase interest and citations.

4) In the introduction, when analyzing prior research, authors make inaccuracies OR provide information that overloads the text, and often their claims are not accompanied by evidence.  It is important for the reader to know the essence of the research you are referring to in your analysis of previous work.

In the introduction, you should analyze previously completed work and note what has been done, what are the shortcomings, what has been done incorrectly.

5) Are you familiar with the work of Mohamad-Hussein, A., & Heiland, J. (2018). 3D finite element modelling of multilateral junction wellbore stability. Petroleum Science, 15(4), 801-814. doi:10.1007/s12182-018-0251-0. How is your study conceptually different from the one presented? 

6) Conclusion is a summary of the research done by the authors, without repetition. This presentation reduces the reader's ease of perception of the information presented. The mistake of incorrect formation of conclusions is a consequence of incorrect presentation of the introduction, noted by me in the remark (5) due to the fact that they did not formulate the goals and objectives when writing the introduction.

The conclusions should briefly characterize the result of the study, for example

As a result of the study

(1) the dependence..... is obtained

(2) it was found that......

(3) and so on.

7) The manuscript has an insufficient list of references (only 24 references). There is no complete coverage of research in terms of geography of citations. The authors refer mostly to their colleagues from China. There are no references to the world experience in this field or related fields, especially to the works of Eastern European, Ukrainian or Russian scientists. 

7.1) It is obligatory to supplement the list of references with researches of scientists from different countries for the last 3-5 years to show geographical (general/global) interest and relevance. 

7.2) When analyzing previous studies, it is not necessary to specify the name of the scientist, for this there is a reference. 

7.3) Authors should avoid group references ".....bedded shale by comprehensively considering 45 interaction between bedding planes and drilling fluids [12-16]".

8) Additional comments. 

- The X and Y axes should be marked on all the graphs.

- Figure 6 and Figure 7 merge into one figure.

- Figure 10 and Figure 11 merge into one figure.

- Figure 14 and Figure 15 merge into one figure

- check the formatting, since there are some layout mistakes in the text. 

Summary: The manuscript is not a finished research paper. Corrections are necessary. The chosen topic of the research is indeed relevant. From my point of view the authors were not able to present their research clearly and competently, which greatly reduced its value and perceived ease of presentation.

From my point of view, the manuscript cannot be published in the open press without correcting the deficiencies indicated in my recommendation

Moderate editing of English language

Round 2

Reviewer 2 Report

The Authors improved this paper really well. I think that this manuscript can by accepted. 

Reviewer 4 Report

The authors have taken into account all my recommendations. From my point of view, the manuscript can be published in the open access

Minor editing of English language required